# Aggressive Variant of Hepatic Epstein–Barr Virus-Associated Inflammatory Pseudotumor-like Follicular Dendritic Cell Sarcoma with PD-L1 and SSTR2a Expression

**DOI:** 10.3390/diagnostics13182916

**Published:** 2023-09-12

**Authors:** Madhur Pardasani, Muthukumarassamy Rajakannu, Mukul Vij, Rajesh Rajalingam, Mohamed Rela

**Affiliations:** Rela Institute & Medical Centre, Chennai 6000044, India; madhurpardasani88@gmail.com (M.P.); muthukumarassamy@yahoo.com (M.R.); rajesh1203@gmail.com (R.R.); mohamed.rela@relainstitute.com (M.R.)

**Keywords:** Epstein–Barr virus, inflammatory pseudotumor-like follicular dendritic cell sarcoma, pathology, programmed death ligand 1, somatostatin receptor 2a

## Abstract

An aggressive Epstein–Barr virus (EBV)-associated inflammatory pseudotumor-like follicular dendritic cell (IPT-like FDC) sarcoma is reported in an adult female. The patient developed multifocal recurrence and passed away 13 months after the initial surgical resection. A bright field microscopic examination of the tumor demonstrated a classical growth pattern and the diffuse expression of Programmed death ligand 1 (PD-L1) and somatostatin receptor 2a (SSTR2a).

**Figure 1 diagnostics-13-02916-f001:**
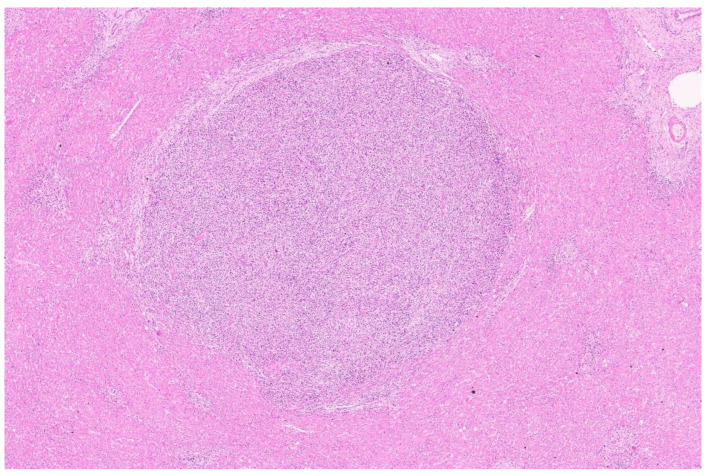
Low-power image displaying a tumor nodule composed of spindle cells admixed with abundant lymphocytes (H&E, ×3). No tumor capsule was noted. The tumor showed an infiltrative pattern in places. The adjacent liver showed mild secondary changes with portal expansion and mild inflammation. A 54-year-old woman presented to us with a persistent biliary fistula (120–150 mL/day) from her right hypochondrial drain. She had undergone a non-anatomic resection of a large tumor on the right lobe of her liver with a thoracoscopic excision of the anterior mediastinal lymph node elsewhere. The histology was reported as an unclassified malignancy with tumor involvement of the surgical hepatic margin. Postoperatively, she developed a high-volume bilious drain output from the cut surface. Endoscopic retrograde cholangiographic stenting was performed 4 months after the initial surgery. The patient was referred to us one month after stenting for a non-healing biliary fistula. She was clinically well, with good performance status and normal liver function. A contrast-enhanced computed tomography (CECT) scan demonstrated two well-organized collections along the cut surface of the right lobe. There were no residual or new lesions in the radiological work-up. In view of the positive previous surgical margin and chronic biliary fistula, completion right hepatectomy was performed at our center. The department of pathology received a hepatectomy specimen weighing 282 g and measuring 10 × 10 × 7.5 cm. The cut surface demonstrated a yellowish-white fibrous area measuring 3.5 × 0.6 cm near the surgical margin. The remaining liver parenchyma was a tan brown color. One tiny, whitish nodule measuring 0.5 cm was also identified in the liver, away from the surgical margin. The sections at the surgical margin demonstrated marked inflammation comprising histiocytes, lymphocytes, plasma cells and polymorphs. The remaining liver showed a largely maintained lobular architecture. Three small tumor nodules measuring 1 mm, 2 mm and 5 mm were identified.

**Figure 2 diagnostics-13-02916-f002:**
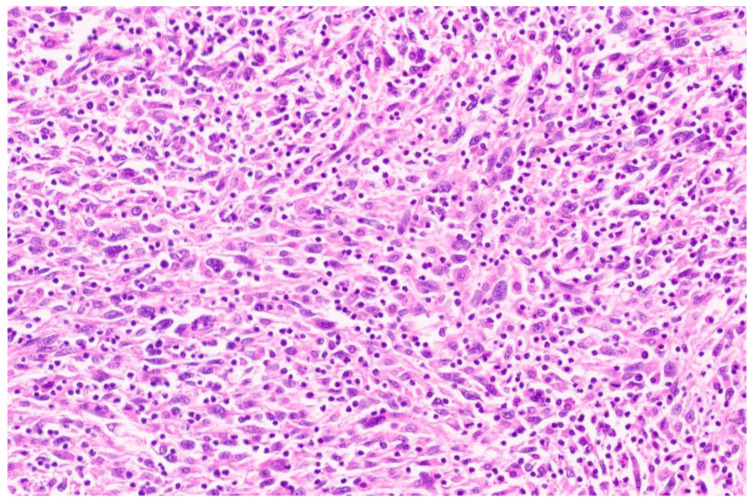
A higher magnification demonstrating spindle-shaped tumor cells arranged in loose fascicles, with oval hyperchromatic nuclei with inconspicuous-to-small nucleoli and moderate eosinophilic cytoplasm with indistinct cytoplasmic borders (H&E, ×20). Mild-to-moderate nuclear pleomorphism was noted. Lymphocytes, histiocytes and a few plasma cells were noted. There was occasional mitosis. No necrosis was noted.

**Figure 3 diagnostics-13-02916-f003:**
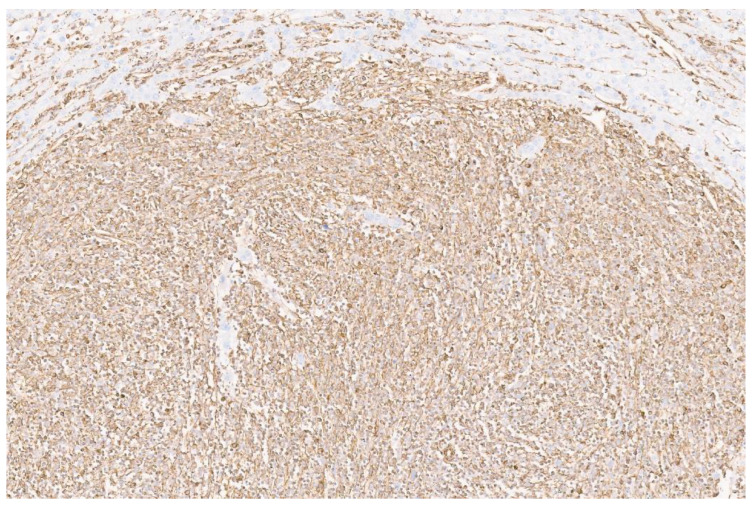
We performed an extensive immunohistochemical study of the tumor cells. The tumor cells were negative for PanCK, EMA, SMA, desmin, s100, SOX-10, myogenin, MYO-D1, INSM1, chromogranin, CD34, ERG, CD117, DOG-1, LCA, ALK-1, STAT-6 and SS18-SSX. The tumor cells showed diffuse positivity for vimentin (Figure 3, ×8).

**Figure 4 diagnostics-13-02916-f004:**
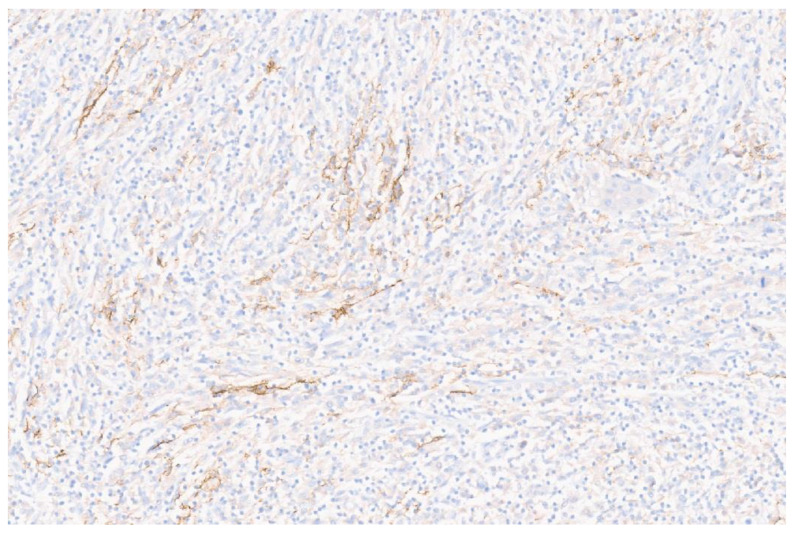
To demonstrate immunophenotype of the follicular dendritic cells (FDC), several immunohistochemical markers were assessed in the tumor. CD23 and CD35 were completely negative. CD21 immunostaining demonstrated patchy positivity in the tumor cells (Figure 4, ×12). Staining of CD21 from focal to diffuse has been reported in the literature [1].

**Figure 5 diagnostics-13-02916-f005:**
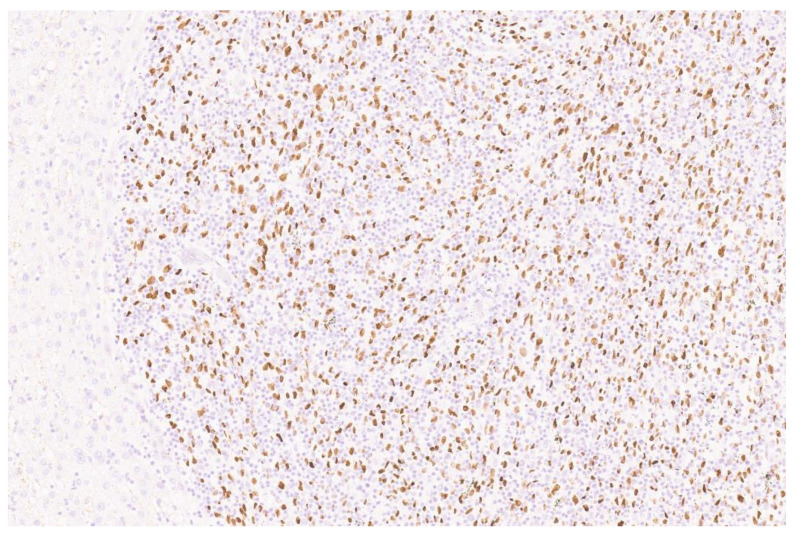
The Epstein–Barr-encoding region (*EBER*) in situ hybridization (ISH) assay showed positive nuclear immunostaining in the spindle cells, whereas inflammatory cells and the surrounded liver parenchyma were completely negative (Figure 5, ×10). With these findings, a final diagnosis of Epstein–Barr virus (EBV) associated inflammatory pseudotumor-like follicular dendritic cell (IPT-like FDC) sarcoma was made. FDC sarcoma is an uncommon mesenchymal tumor that occurs in lymph nodes or in extranodal sites, including the tonsils, gastrointestinal tract, lung, mediastinum and peritoneum [2]. Conventional FDC sarcoma occurs across a wide age range, with a peak age of around 50 years at presentation [3]. There is no gender predilection. Morphologically, FDC sarcomas are composed of ovoid-to-spindle-to-epithelioid neoplastic cells that are arranged in whorls, fascicles, syncytial sheets, nodules and a storiform pattern [3]. An admixed mild infiltrate comprising of small lymphocytes is also noted. In general, FDC sarcoma is a low- to intermediate-grade malignant tumor, and up to half of patients die of the disease in long-term follow-up [4]. There is a rare subgroup of FDC sarcoma which is EBV--driven and is called IPT-like FDC sarcoma [1]. These tumors usually show predilections for the liver and spleen and demonstrate distinctive clinical and pathological features, such as a marked female predilection, indolent clinical course, the presence of a prominent inflammatory background and an association with EBV [5]. EBV-related IPT-like FDC sarcoma presents at a median age of 52.5 years and it is most common in Asians, possibly reflecting endemic EBV infections in this region [4]. The pathogenetic mechanism of EBV-related IPT-like FDC sarcoma is not clear, but EBV infection is a significant etiological factor as the tumor cells are consistently associated with clonal EBV genomes [6]. It has been suggested that the tumor may arise from a common EBV-infected mesenchymal cell that differentiates along the follicular or dendritic cell pathways. Li et al. recently reported 13 cases of EBV-related IPT-like FDC sarcoma [1]. The authors also reviewed 70 published cases. The tumors occurred mainly in the liver and/or spleen, followed by the colon, with a few cases involving the lungs, tonsils, adenoid, peripancreas and upper abdomen. They reported three growth patterns on the basis of the degree of lymphoplasmacytic inflammatory cell infiltration and the characteristics of the blood vessels within the tumors: classic type, lymphoma-like subtype and hemangioma-like subtype [4]. The classic subtype shows distinct spindle-shaped tumor cells which are arranged in fascicles or a storiform pattern with abundant lymphoplasmacytic infiltrates. Our case showed the classic growth pattern. The lymphoma-like subtype shows singly dispersed neoplastic cells admixed with marked lymphoplasmacytic infiltrates. This subtype can easily be misdiagnosed as low-grade B-cell lymphoma. The tumor cells are difficult to identify even in a high-power field. The hemangioma-like growth pattern shows prominent vasculature in various shapes and sizes, resembling a cavernous hemangioma.

**Figure 6 diagnostics-13-02916-f006:**
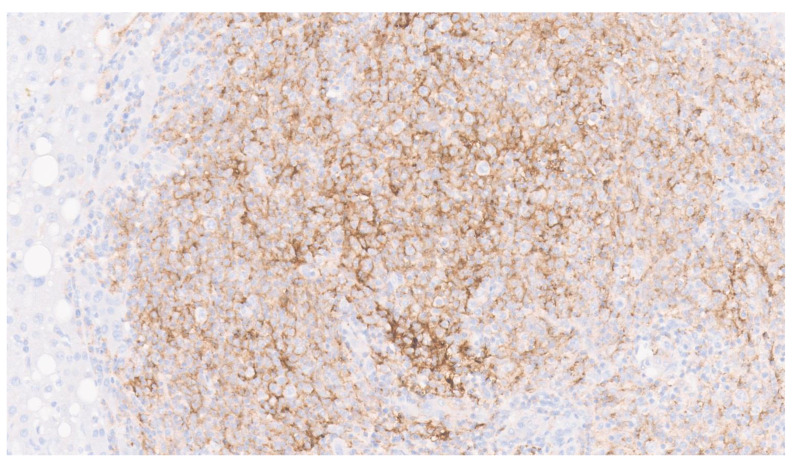
Programmed death ligand 1 (PD-L1) immunostaining showed diffuse, strong membranous positivity in both tumor cells and admixed immune cells (Figure 6, ×15). PD-L1 is a 290-amino acid transmembrane glycoprotein. It is expressed on multiple cell types, including antigen presenting cells (APCs), T and B cells, monocytes and some epithelial cells, particularly under inflammatory conditions [7]. The expression of PD-L1 has also been demonstrated in various malignant tumors as a mechanism to escape the anti-tumor immune response [7,8]. PD-L1 expression has also been suggested as a predictive biomarker of response to anti PD-L1 immunotherapies and may be utilized in recurrent, metastatic or inoperable cases of EBV-related IPT-like FDC sarcoma. PD-L1 expression was also demonstrated in all cases (range: 5–90%) of EBV-related IPT-like FDC sarcoma by Li et al. [1].

**Figure 7 diagnostics-13-02916-f007:**
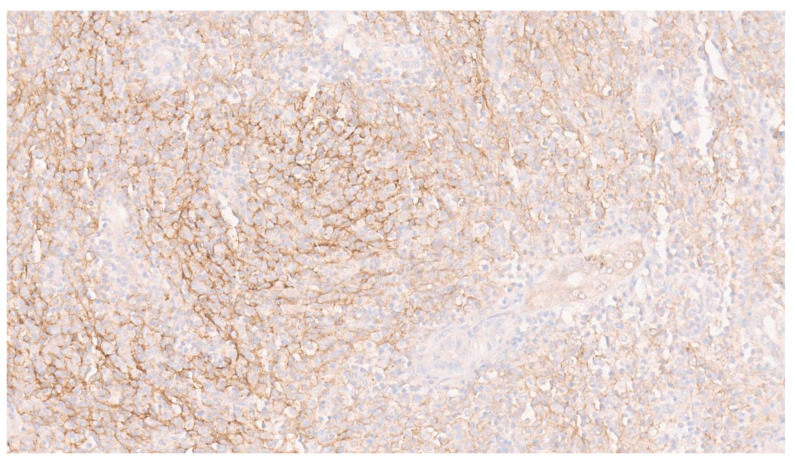
Somatostatin receptor 2a (SSTR2a) was diffusely and strongly positive in the tumor (Figure 7, ×8). SSTR2a has been reported as a robust marker of FDC and was shown to be positive in FDC sarcomas [9,10]. Recently, it has also been reported to be positive in EBV-associated IPT-like FDC sarcoma [1]. Somatostatin (SST) is an inhibitor of hormone secretion that is widely expressed in multiple organs. SST exerts its biological function through its inhibitory G-coupled protein receptor, SSTR. SSTR has 5 subtypes (SSTR1-5). SST performs proapoptotic, antiangiogenic and antiproliferative functions after activation [9]. SSTR2a is expressed specifically in neuroendocrine tumors [11]. The immunostaining of SSTR2a has also been reported in multiple non-neuroendocrine tumors, including meningioma, synovial sarcoma and phosphaturic mesenchymal tumors, and rarely in solitary fibrous tumors [9,10,12]. No cranial or spinal tumor was detected in our patient. Our tumor was negative for EMA, STAT-6 and SSX-XX18 and showed patchy expression of CD21, another immunomarker for FDCs. Anther differential diagnosis of EBV-related IPT-like FDC sarcoma is a diagnosis of an inflammatory myofibroblastic tumor (IMFT); however, IMFTs are negative for CD21, CD35 and EBER ISH. ALK-1 was also negative in our tumor. In a series of cases of EBV-related IPT-like FDC sarcomas, immunohistochemical workups showed variable staining for FDC markers (CD21, CD23, CD35 and SSTR2a) [1]. Considering the inconsistent expression of different FDC markers in the neoplastic cells, performing immunohistochemical staining for multiple FDC immunomarkers has been suggested to document the FDC nature of the neoplastic cells. We also demonstrated the expression of SSTR2a and suggest that this marker can also be used to document the FDC nature of a tumor.

**Figure 8 diagnostics-13-02916-f008:**
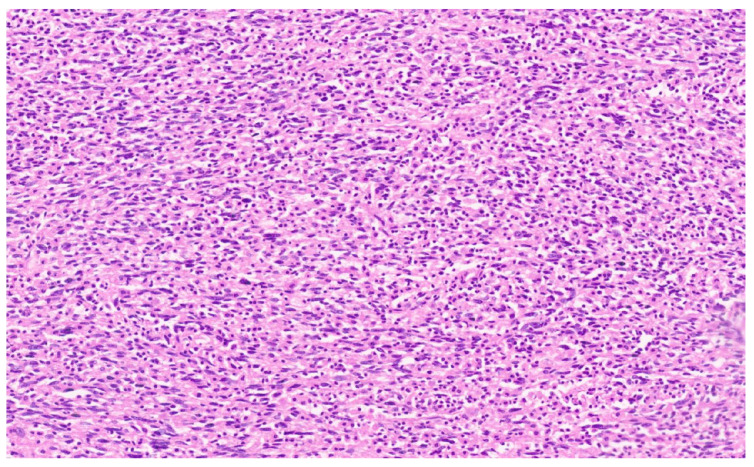
A review of the tissue blocks from the previous surgery showed a similar tumor, with sheets of spindle cells with oval-to-elongated mildly anisomorphic hyperchromatic nuclei and an eosinophilic cytoplasm admixed with lymphomonuclear cells (Figure 8, H&E, ×15). CD3-, CD20- and CD68-positive immune cells were identified in the tumor. The patient had no associated immunodeficiency syndromes. An immunoassay for EBV IgG antibodies to viral capsid antigen (VCA) and Epstein–Barr nuclear antigen (EBNA) was performed. It was negative. An EBV polymerase chain reaction (amplicon: the 97 bp region of the EBNA-1 gene) was also negative. Whole-exome sequencing using blood did not reveal any significant mutation. Multifocal recurrence was observed in the liver, lungs and mediastinum at 3 months after the second operation and 8 months after the first surgery. No further treatment was offered to her. She died 13 months after the first surgery. EBV-related IPT-like FDC sarcomas have a low to intermediate grade of malignancy. Li et al. reported tumor recurrence in two patients, and all thirteen patients in his series were alive with a median follow-up of 28.5 months [1]. This is in contrast to our case, in which the tumor showed highly aggressive features and the patient died 13 months post-surgery. To conclude, we report a highly aggressive variant of EBV-related IPT-like FDC sarcoma in an adult female which showed the expression of PD-L1 and SSTR2a. The expression of PD-L1 suggests that immunotherapy may be offered in inoperable, recurrent or metastatic disease; however, further investigation is needed in this regard.

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
