# Peer review of "Aggressive Variant of Hepatic Epstein–Barr Virus-Associated Inflammatory Pseudotumor-like Follicular Dendritic Cell Sarcoma with PD-L1 and SSTR2a Expression"

_diagnostics, 2023, doi:10.3390/diagnostics13182916_

Round 1
Reviewer 1 Report
The Author describe a very rare form of IPT-like FDC sarcoma with aggressive clinical course,
The morphology and the presence of EBV are consistent with this diagnosis , however the Authors should better describe the possible differential diagnosis in relation also to the staining of SSTR2a which has been reported as a marker of FDC but negative in IPT-like FDC sarcoma.
The Authors should report uniformly in the text SSTR2a
minor editing of english
Author Response
Dear Reviewers and Editors,
Many thanks for your valuable comments regarding our article entitled: Aggressive variant of Hepatic Epstein-Barr virus associated inflammatory pseudotumor-like follicular dendritic cell sarcoma with PD-L1 and SSTR2a expression.. We appreciate your interest and precious time spent in going through our article. We have now revised our manuscript considering the comments, critiques and questions highlighted in your reviews. We believe that the revised manuscript now reads well and fulfills the requirements for publication in Diagnostics. If you have any further queries or comments, please do not hesitate to contact us.
Kind Regards
Thank you
Mukul vij
Reviewer no 1:
The Author describe a very rare form of IPT-like FDC sarcoma with aggressive clinical course, The morphology and the presence of EBV are consistent with this diagnosis , however the Authors should better describe the possible differential diagnosis in relation also to the staining of SSTR2a which has been reported as a marker of FDC but negative in IPT-like FDC sarcoma.
Reply: SSTR2a has been reported as marker of follicular dendritic cells and was shown to be in positive in follicular dendritic cell sarcomas. Recently it has also reported to be positive in Epstein-Barr virus associated inflammatory pseudotumor-like follicular dendritic cell sarcoma. Immunostaining of SSTR2a has been reported in neuroendocrine and non-neuroendocrine tumours with spindle cells including meningioma, synovial sarcoma and rarely in solitary fibrous tumour. No cranial or spinal tumor was detected in our patient. Our tumour was negative for EMA, STAT-6 and SSX-XX18 and showed expression of CD21, another marker for follicular dendritic cells. Expression of EBER-ISH excluded all other possibilities including inflammatory myofibroblastic tumor, and inflammatory pseudotumor.
The Authors should report uniformly in the text SSTR2a.
Reply: Corrections are done in the manuscript.
Reviewer 2 Report
The paper, which is classified as an interesting images and presents a presentation of a relatively rare tumor associated with EBV, is infact a case report or study case. However, in addition to the images of the pathohistological findings and the obtained immunohistochemistry, which logically explain the presented case, the work is not well organized. I think that it should be composed according to the case report, it should be summarized textually, more information about the patient should be given, and the pictures of the pathohistological findings should be like an appendix. In particular, a whole discussion is conducted under Figure 7. Short explanations should go below the pictures, not the text of the introduction of this case presentation (Figure 1., eg.) or the already mentioned discussion (below Figure 7., eg).
Also, when the authors talk about the examined serological EBV status of the patient, which is negative, they do not go into detail by explaining which antibodies were checked by which immunoenzymatic tests. The very complex antigenic composition of EBV is known, and testing the serological status requires a very wide range of EBV antibody tests. The authors of the paper only report EBV negative??? Also, was another molecular method performed by checking the patient's blood for a set of latent EBV genes, was it done?
Check the English language as well.
Author Response
Dear Reviewers and Editors,
Many thanks for your valuable comments regarding our article entitled: Aggressive variant of Hepatic Epstein-Barr virus associated inflammatory pseudotumor-like follicular dendritic cell sarcoma with PD-L1 and SSTR2a expression.. We appreciate your interest and precious time spent in going through our article. We have now revised our manuscript considering the comments, critiques and questions highlighted in your reviews. We believe that the revised manuscript now reads well and fulfills the requirements for publication in Diagnostics. If you have any further queries or comments, please do not hesitate to contact us.
Kind Regards
Thank you
Mukul vij
Reviewer no 2:
The paper, which is classified as an interesting images and presents a presentation of a relatively rare tumor associated with EBV, is infact a case report or study case. However, in addition to the images of the pathohistological findings and the obtained immunohistochemistry, which logically explain the presented case, the work is not well organized. I think that it should be composed according to the case report, it should be summarized textually, more information about the patient should be given, and the pictures of the pathohistological findings should be like an appendix. In particular, a whole discussion is conducted under Figure 7. Short explanations should go below the pictures, not the text of the introduction of this case presentation (Figure 1., eg.) or the already mentioned discussion (below Figure 7., eg).
Reply: We have revised the manuscript. The report is better organized now. Descriptions are added below each image.
Also, when the authors talk about the examined serological EBV status of the patient, which is negative, they do not go into detail by explaining which antibodies were checked by which immunoenzymatic tests. The very complex antigenic composition of EBV is known, and testing the serological status requires a very wide range of EBV antibody tests. The authors of the paper only report EBV negative??? Also, was another molecular method performed by checking the patient's blood for a set of latent EBV genes, was it done?
Reply: Immunoassay for IgG antibody to viral capsid antigen (VCA IgG) and Epstein-Barr nuclear antigen (EBNA) was done. It was negative. EBVPCR (97 bp region of the EBNA-1 gene) was also negative.
Round 2
Reviewer 2 Report
After the corrections have been made, the paper is better organized, easier and more adequate for follow-up. The entered corrections correlate with the correction in the citation of references.